# Vaccination against Rabbit Hemorrhagic Disease Virus 2 (RHDV2) Using a Baculovirus Recombinant Vaccine Provides Durable Immunity in Rabbits

**DOI:** 10.3390/v16040538

**Published:** 2024-03-30

**Authors:** Angela M. Bosco-Lauth, Amber Schueler, Edward Midthun, Hailey Tyra, Amanda Held, Claire Hood, Marissa Quilici, Sara Erickson, Sherry Glover, Bradley Gustafson, Gary Anderson

**Affiliations:** 1Department of Biomedical Sciences, Colorado State University, Fort Collins, CO 80523, USAhailey.tyra@colostate.edu (H.T.);; 2Medgene, Brookings, SD 57006, USA

**Keywords:** rabbit hemorrhagic disease virus 2, RHDV2, vaccine, domestic rabbit

## Abstract

Rabbit hemorrhagic disease virus 2 (RHDV2) emerged in the United States in 2018 and has spread in both domestic and wild rabbits nationwide. The virus has a high mortality rate and can spread rapidly once introduced in a rabbit population. Vaccination against RHDV2 provides the best protection against disease and should be considered by all rabbit owners. Here, we investigate the duration of immunity provided by vaccination with the Medgene Platform conditionally licensed commercial vaccine 6 months following the initial series. Rabbits received either the vaccination or a placebo and were challenged with RHDV2 6 months later. All vaccinated rabbits survived challenge whereas 18/19 non-vaccinated controls succumbed to infection within 10 or fewer days post-challenge. These results demonstrate lasting immunity following vaccination with the Medgene RHDV2 vaccine.

## 1. Introduction

Rabbit hemorrhagic disease (RHD) is a viral disease of rabbits caused by a virus in the caliciviridae family (genus *Lagovirus*) [1]. The first outbreak of RHD was documented in China in 1984 and in less than one year killed over 140 million domestic rabbits [2,3]. Over the next several decades, outbreaks occurred in Europe, North Africa, and the Americas [1]. In New Zealand and Australia, where rabbits are considered an agricultural pest, the virus was released intentionally as a means of biological population control, where it reduced the population by nearly 95% [4]. In 2010, a novel strain of the virus, referred to as RHDV GI.2 or RHDV2, emerged in France and has since spread globally to nearly every continent, including Europe, Asia, Australia, Africa, and North America [5,6,7,8,9]. The original virus responsible for RHD, now called RHDV G1.1 or RHDV, has a high mortality rate in European rabbits (*Oryctolagus cuniculus*) but typically does not cause disease in other lagomorph species, while RHDV2 has a much broader host range, with documented infections in a variety of wild lagomorphs, including cottontails (*Sylvilagus* spp.) and hares (*Lepus* spp.) [10,11]. With the emergence and convergence of both viruses in much of the world, the economic impact on the rabbit industry as well as the threat posed to wild lagomorph species is considerable. 

Both RHDV and RHDV2 cause disease characterized by hemorrhage and sudden death, with mortality rates between 60 and 90% depending on age and species. However, RHDV2 has a tendency to cause higher mortality in young rabbits compared to RHDV [12]. RHDV2 has become the dominant genotype circulating globally, but RHDV still exists, and cross-protection between the two strains is minimal [12]. Culling of rabbits is one option to slow the spread of disease, but so far only Mexico has been able to eradicate an outbreak of RHDV in this manner [13]. The broader host range of RHDV2 into wild lagomorphs, particularly those of conservation concern, makes eradication via culling both unappealing and impractical. Therefore, strict biosecurity measures and vaccination remain the most promising options for mitigation and control of the disease. 

There are currently two licensed inactivated vaccines against RHDV2 in Europe: Filavac and Eravac [14,15]. Other RHDV and RHDV2 vaccines, including the recently licensed YURVAC-RHD, have largely targeted the capsid protein, VP60, a major viral structural and immunogenic protein, and have either expressed this protein in a recombinant viral vector or used virus-like particles [16,17,18,19,20,21]. Many of these vaccines and vaccine candidates are bivalent or even trivalent, covering RHDV, RHDV2, and rabbit myxoma virus, a poxvirus that also causes fatal disease in rabbits [22,23,24,25]. In the United States, only RHDV2 is endemic, and therefore monovalent RHDV2 vaccines are appropriate. Because lagoviruses do not grow in cell culture, inactivated viral vaccines require harvesting viral-laden tissue from rabbits infected with live virus and rendering the virus inactive, thereby requiring high containment and significant numbers of animals. By contrast, recombinant technology does not require that the virulent virus to be passaged and harvested from animals, making this option both safer and more efficient [15]. Previously, we demonstrated that a baculovirus expressing VP60 provides protection against challenge with RHDV2 in rabbits following a two-dose prime-boost regimen [26]. Consequently, that vaccine has been available for emergency use for high-risk rabbits and conditionally licensed in the U.S. Here, we demonstrate that the same vaccine dose schedule protects rabbits from lethal infection 6 months following vaccination. 

## 2. Materials and Methods

### 2.1. Animals

A total of 55 New Zealand white rabbits (*Oryctolagus cuniculus*), approximately 7 weeks of age and of roughly equal male/female ratio, were obtained from a Specific and Opportunistic Pathogen Free (SOPF) production colony and assessed for general health prior to enrollment into randomly assigned blinded treatment groups. Duration of the study, animal use reduction, and requirements to achieve statistical significance for licensure of the product were taken into consideration when determining enrollment numbers. Rabbits were naïve to RHDV2 prior to enrollment. Rabbits received either a full series (two doses) vaccination against RHDV2 or two doses of placebo. During the vaccination phase, rabbits were individually housed in 29.27”W × 28.19”D x 18.55”H cages in racks of three within a common room. Rabbits were fed alfalfa pellets supplemented with Timothy hay cubes and carrots. Water was available ad libitum. All procedures during the vaccination phase were performed in accordance with Medgene IACUC approval (#22-005). Blood was collected prior to first vaccination (Study Day 0), 21 days post-first vaccination, 91 days post-first vaccination, and prior to challenge at 226 or 227 days post-first vaccination. Six months following vaccination on SD220, 45 rabbits were transported to Colorado State University (CSU), while the remaining rabbits were maintained and boosted at 12 months to determine antibody response at 12 months and post-annual booster; these rabbits were not challenged with RHDV2. One rabbit was removed from the study prior to the 12-month booster for humane reasons unrelated to vaccination. Once at CSU, rabbits were individually housed in 27”W × 27”D × 17.7”H cages in banks of 6 cages in one of two identical rooms in the animal biosafety level 3 (ABSL3) facility during the challenge phase. Rabbits were provided ad libitum water and feed consisting of commercial alfalfa pellets supplemented with grass hay, carrots, and apples. All procedures at CSU were performed in accordance with University IACUC approval (#1161).

### 2.2. Vaccine Preparation

The vaccine being tested is a commercially available inactivated (killed) baculovirus-derived recombinant subunit vaccine, directed at eliciting an immune response to the immunogenic VP60 protein of US strains of RHDV2 (patent pending). The complete nucleotide sequence for the VP60 protein has been previously published [26] and was obtained from the United States Department of Agriculture (USDA) Animal and Plant Health Inspection Service (APHIS) from the circulating strain in 2020. The recombinant VP60 is adjuvanted with aluminum hydroxide to further stimulate the immune response in this proprietary product. An adjuvant-matched placebo lacking antigenic proteins was also prepared for use in this study.

### 2.3. Vaccine Administration

On SD0 and SD21, 55 rabbits were vaccinated subcutaneously with a 0.5 mL dose of either the commercially available test vaccine (*n* = 30) or the adjuvant matched placebo (*n* = 25). Enrollment to either test group was random, and the test product was blinded to all involved in the execution of the study until conclusion.

### 2.4. Virus

Challenge material originated from livers from RHDV2-naturally infected rabbits during the 2020–2021 U.S. outbreak and supplied by the United States Department of Agriculture (USDA). Challenge material was transferred to the Animal Disease Laboratory (ADL) at Colorado State University (CSU), a large animal Biosafety Level 3 facility (BSL3). Livers were pooled and homogenized in phosphate-buffered saline (PBS) at a 1:1000 ratio for the starting challenge material. 

### 2.5. Challenge

Following the 6-month vaccination phase, on SD220, a total of 45 rabbits were transported to CSU and housed according to a randomized assignment such that CSU study participants were blinded to the study groups. Vaccinates and placebo controls, mixed sex, were roughly equal in each room. Rabbits were allowed to acclimate for 7 days, during which time they were subcutaneously implanted with biothermal microchips, and a baseline blood sample was collected. Two sentinel rabbits were euthanized and necropsied during the acclimation time, and livers were harvested for PCR analysis to confirm lack of RHDV2 exposure. On day 7 post-arrival (SD228 post-vaccination), the remaining 43 rabbits were challenged orally with 1 mL of a 1:1000 RHDV2 liver homogenate using a 3 mL syringe with a blunt stainless steel gavage needle attached. The undiluted liver homogenate had a CT value of 13.981, while the 1:1000 diluted inoculum had a CT of 21.621. Following challenge, rabbits were monitored at least twice and up to four times daily for signs of clinical disease, and body temperatures were recorded daily. Animals that succumbed to infection or that were euthanized via pentobarbital overdose due to endpoint criteria (moribund, anorexic > 2 days, dyspneic, hemorrhagic discharge from nose or mouth) were necropsied, and livers were saved for RT-PCR analysis. All other rabbits were euthanized on day 10 post-infection (SD238 post-vaccination), and livers were harvested for RT-PCR analysis.

### 2.6. RT-PCR Analysis

Challenge material and livers from all rabbits were tested for presence of RHDV2 RNA by real-time Taqman polymerase-chain reaction (RT-PCR), as previously described [27]. Briefly, livers were prepared for extraction by homogenizing in lysis buffer using Qiagen RNeasy™ extraction kits, as per the manufacturer’s instructions (Qiagen, Hilden, Germany), and RT-PCR was performed using TaqMan™ Fast Virus 1-step Mastermix kit (Thermo Fisher Scientific, Waltham, MA, USA).

### 2.7. Serology

An indirect ELISA-utilizing recombinant baculovirus produced RHDV2 VP60 protein as capture was utilized for the assessment of serological response. Briefly, plates were coated overnight with 0.1 µg per well of rBaculovirus-RHDV2 protein diluted in carbonate coating buffer. Coated plates were washed 3× with 0.05% Tween in 1× PBS wash buffer and then blocked with a 1% BSA/10% FBS/0.05% PBST blocking buffer for 2 h at 37 °C. Plates were washed 3× with the aforementioned wash buffer. Two-fold serial dilutions of the test sera, polyclonal primary antibody collected from the rabbits enrolled in the study starting at a 1:200 dilution in PBS, were performed and transferred to duplicate wells of the blocked plate. The test sera dilutions were allowed to incubate at 37 °C for 1 h. Plates were washed 3× with the aforementioned wash buffer. Goat Anti-Rabbit Horse Radish Peroxidase (HRP) was diluted 1:10,000 in PBS and added to the plates. Plates were allowed to incubate at 37 °C for 1 h. Plates were washed 3× with the aforementioned wash buffer. Peroxidase substrate was added to the plates, and plates were allowed to develop at room temperature for 8 min. 1N sulfuric acid was added to the plates to stop the reaction. Plates were read for absorbance at 450 nm. Data were analyzed utilizing a 0.222 optical density cutoff, which was previously established with SPF rabbit sera run on multiple days, with multiple scientists incorporating 3 standard deviations from the average. The inverse reciprocal of the dilution for each sample was reported. 

This methodology was utilized in testing the serological response of 20 rabbits (10 from each treatment group) during the vaccination phase (SD0 and SD91) and of all rabbits at 7 months post-second vaccination (SD226 or 227). The described method was also used on 9 remaining rabbits that were not transported to CSU and withheld from the challenge phase. The 9 rabbits (representing both treatment groups) were followed serologically for 12 months post-second vaccination, receiving a single-dose booster of the vaccine at 12 months post-second vaccination, with additional serology at 14 and 28 days post-booster.

## 3. Results

### 3.1. Challenge Results

Of the 43 RHDV2-challenged animals, 18 of the 19 non-vaccinated animals succumbed to infection and were either found dead or euthanized between days 2 and 7 post-infection. The majority (16/18) succumbed between days 2 and 3, with one rabbit euthanized on day 6 and another on day 7 post-infection. The most common clinical sign observed prior to death was fever (>40.2 °C); other signs included lethargy, anorexia, and weakness. It is notable that fever was not observed in all rabbits that ultimately succumbed to infection. The remaining 25 rabbits (1 control and 24 vaccinates) were subclinical throughout the challenge phase. At necropsy, the most common gross finding across all clinical rabbits was generalized systemic hemorrhage in which multiple organs (lung, liver, spleen, kidney) were involved and free fluid, typically blood-tinged, was found in the abdomen. 

### 3.2. PCR Analysis

Liver samples were collected from all rabbits during necropsy and tested for presence of RHDV2 viral RNA by RT-PCR. All rabbits that succumbed to infection had RT-PCR-positive livers, with CT values ranging from 12.8 to 17.5 (Table 1). By contrast, in all but two of the surviving rabbits, RHDV2 RNA was undetectable by PCR, and the two that were positive had CT values of 25.7 and 33.9 (Table 1). The sole unvaccinated rabbit had the lower of those two scores, indicating infection that was resolving. Neither of the sentinel rabbits had detectable viral RNA in their livers. 

### 3.3. Serology

A random selection of 10 rabbits from each treatment group, vaccinate and placebo, were bled on SD0 and SD91 during the vaccination phase. Prior to challenge, all rabbits enrolled had a blood sample collected on SD226 or SD227. The blood was processed for serum and held for concurrent testing of all samples. All 20 rabbits had RHDV2 VP60 titers equal to or less than the cutoff of 200 on SD0. The 10 rabbits from the vaccinate group exhibited titers of ≥1600 on SD91 following both doses of vaccine, while the placebo group maintained titers lower than the cutoff of 200 (Table 2). Just prior to challenge, all rabbits receiving the placebo had titers below cutoff, while the vaccinate group had geometric mean titer of 1600, although titers did decrease between SD91 and SD226/227 for most of the rabbits sampled at those times. (Table 3)

The 9 rabbits (5 placebo, 4 vaccinates) that were held for serological assessment beyond the 6-month duration of immunity had sera collected at 7 months post-second vaccination (SD226), 9 months post-second vaccination (SD296), and at 12 months post-second vaccination (SD388). Rabbits receiving the placebo maintained titers <200, and the geometric mean of the 4 vaccinated rabbits decreased from 1902 at 7 months to 951 at 12 months. These 9 rabbits were then administered a single dose of vaccine following the 12-month collection (SD388/0DPB). Blood was collected 14 days and 28 days post-booster (DPB). Following the single-dose administration, the vaccinates illustrated a strong booster response with geometric mean titers jumping to 18,101 and 15,221 on 14 and 28 days post-booster, respectively. Four of the five rabbits that received a single dose seroconverted, while one did not; all rabbits that received 2 doses of vaccine seroconverted (Table 4).

## 4. Discussion

Rabbit hemorrhagic disease virus 2 has spread to 5 continents and has been confirmed in 29 states within the U.S. in addition to two Canadian territories at the time of this publication [28,29]. The disease is characterized by high mortality (60–90%) in domestic and European rabbits and can infect wild lagomorph species with variable results. The economic impact on the rabbit meat industry is significant, not to mention the impact on pet trade and wildlife species of conservation concern. Rabbits are the 3rd most common companion mammal behind dogs and cats in the U.S., and as of 2017, account for roughly half a million food animals on more than 4000 farms [30]. Rabbits are also raised for show exhibition, manure production, fur/pelts, and as an alternative meat for pet food. Other countries consider rabbit meat a mainstay as a food source, including China, Korea, and much of Europe. In addition to pets and farm animals, rabbit hunting is a common practice worldwide, and in the U.S., approximately 1.3 million people hunt rabbits each year, contributing to the roughly $1.6 billion revenue generated by small game hunters [31]. Therefore, the need to manage and minimize the impact of RHDV2 cannot be overstated. 

Vaccines against RHDV2 have been in existence since 2016 and have been highly efficacious in preventing disease for vaccinated animals [15]. In places where RHDV and RHDV2 cocirculate, inactivated multivalent vaccines like Filavac are deployed [15]. However, in the U.S., only RHDV2 is endemic and as such, vaccines can target this genotype specifically. The baculovirus-vectored recombinant vaccine produced by Medgene has been shown to be highly efficacious in preventing disease, and the current study not only confirms this efficacy, but also demonstrates lasting immunity over 6 months post-vaccination. Indeed, other baculovirus-vectored RHDV2 vaccines have shown similar results, with immunity lasting up to 14 months in most individuals [21], suggesting that this vaccine platform elicits strong humoral and cell-mediated immunity. In this study, antibody titers coupled with a strong protective response against infection suggest that humoral immunity is highly indicative of a protective response. Furthermore, antibody titers remained at the level of protection for a full 12 months, with a dramatic increase in titers following a booster, suggesting that annual boosters would provide a robust increase in immune response and are highly likely to confer lasting immunity. Interestingly, one rabbit (#677) that received only a single dose of the vaccine failed to seroconvert, while all rabbits receiving a prime-boost series developed a strong antibody response, thereby indicating that a two-dose series is ideal for optimum response. In the prior study {26}, vaccination prevented disease but did not prevent infection, as was confirmed by the presence of RHDV2 RNA in the livers of vaccinated and infected animals 10 days post-infection; but in the current study, all but one vaccinated rabbit was able to completely clear viral RNA from the liver by 10 days post-infection. The major difference between these two studies, in addition to time between vaccination and challenge, is that the first study utilized group housing of vaccinates and controls, so it is possible that vaccinated animals were continually exposed to infectious virus material shed by control animals into the environment and therefore received multiple “inoculations” during the course of the study. RHDV can persist in the environment and maintain infectivity for at least 91 days, and viral RNA can also persist in animals that survive infection for 3 months, so it is unsurprising to find evidence of infection in these vaccinated animals [32,33]. Importantly, in the current study, none of the vaccinates displayed any signs of clinical disease, while 95% (18/19) of the placebo-vaccinated controls died or were euthanized due to severe clinical disease during this same time frame. These results clearly demonstrate that vaccination is highly effective in preventing disease and disease-associated mortality. 

Because RHDV2 can persist in the environment, and because vaccination does not necessarily prevent infection and subsequent shedding of infectious virus, the only way to prevent the spread of this virus within a rabbit facility is to ensure that all rabbits are vaccinated. Based on the PCR results from the livers of vaccinated-infected animals, it is likely that vaccinated animals shed less virus and for a shorter period of time than animals that are infected and recover, but since we did not specifically test rabbit feces or other bodily fluids for presence of infectious virus, we cannot confirm that vaccinates are not shedding. Previous studies show that inoculating rabbits with fecal material from RHDV2-infected rabbits can result in disease, so it is likely that rabbits can acquire infection from coming into contact with feces or other material from infected rabbits [32]. Furthermore, lagoviruses are extremely hardy and can persist in the environment on feces and in infected tissue for months [34,35], so current biosecurity measures need to consider that bedding and cages are infectious unless decontaminated using bleach or other proven methods of inactivation. This is particularly important for rabbits who attend shows or fairs and are housed in contact with other rabbits or their bedding. However, all rabbits that are housed outdoors or in any facility where they may encounter wild rabbits would benefit from vaccination. It is unlikely that RHDV2 is eradicable from the U.S. or any country where wild rabbits have been infected, and vaccination of wild rabbits is not a feasible option for controlling the spread of the virus, so ultimately the burden rests on rabbit owners to mitigate this risk. Future studies should focus on whether or not vaccinated animals are capable of shedding infectious virus following exposure to RHDV2 and should characterize the duration of shedding of infectious material. At present, the most viable option for preventing disease is to vaccinate all individuals at risk of exposure.

## 5. Conclusions

Vaccination of domestic rabbits using the commercially available Medgene RHDV2 vaccine provides lasting immunity and prevents disease in animals beyond 6 months post-vaccination. Rabbit owners with animals at risk of RHDV2 exposure are encouraged to consider vaccination of their animals as a primary source of disease prevention. 

## 6. Patents

Patent Pending.

## Figures and Tables

**Table 1 viruses-16-00538-t001:** CT values from rabbit livers harvested at time of death or euthanasia.

Rabbit ID	Vaccinated	Day Post-Infection	CT Value ^1^
F04 *	No	−2	Undetected
354 *	Yes	−2	Undetected
A94	Yes	10	Undetected
C2A	Yes	10	Undetected
F8B	Yes	10	Undetected
95E	Yes	10	Undetected
0AF	Yes	10	Undetected
1A4	Yes	10	Undetected
E73	Yes	10	Undetected
06C	Yes	10	Undetected
179	Yes	10	Undetected
537	Yes	10	Undetected
A8C	Yes	10	Undetected
056	Yes	10	33.914
A17	Yes	10	Undetected
934	Yes	10	Undetected
8E5	Yes	10	Undetected
361	Yes	10	Undetected
BFD	Yes	10	Undetected
A03	Yes	10	Undetected
F85	Yes	10	Undetected
C80	Yes	10	Undetected
EF2	Yes	10	Undetected
48A	Yes	10	Undetected
20D	Yes	10	Undetected
CDE	Yes	10	Undetected
3BC	No	10	25.651
267	No	2	16.468
B2E	No	2	14.470
DDB	No	2	14.963
D4E	No	2	14.971
0D7	No	2	13.798
FFC	No	2	13.902
765	No	2	13.776
796	No	2	15.530
EC3	No	2	12.844
D81	No	3	14.005
196	No	3	13.449
972	No	3	13.847
E8B	No	3	15.723
35D	No	3	17.104
E59	No	3	13.609
2C6	No	3	17.508
F91	No	6	14.671
6DA	No	7	15.198
Inoculum			21.621

* Non-inoculated sentinel rabbits euthanized prior to challenge. ^1^ CT values of 35 or greater were considered negative and recorded as undetected.

**Table 2 viruses-16-00538-t002:** Serological response from the rBaculovirus-derived RHDV2 VP60 ELISA for the rabbits challenged with RHDV2.

.		Study Day					Study Day		
Group	Rabbit ID	0	91	226 or 227	Group	Rabbit ID	0	91	226 or 227
1—Vaccinate	056	.	.	1600	2—Placebo	196	.	.	<200
	179	<200	3200	1600		267	.	.	<200
	354	.	.	800		765	<200	<200	<200
	361	.	.	1600		796	.	.	<200
	537	.	.	1600		972	<200	<200	<200
	934	.	.	800		0D7	<200	<200	<200
	8E5	.	.	3200		2C6	.	.	<200
	06C	.	.	800		35D	.	.	<200
	0AF	.	.	1600		3BC	<200	<200	<200
	1A4	<200	6400	3200		6DA	<200	<200	<200
	20D	.	.	3200		B2E	.	.	<200
	48A	.	.	3200		D4E	<200	<200	<200
	95E	.	.	1600		D81	<200	<200	<200
	A03	200	6400	1600		DDB	<200	<200	<200
	A17	.	.	3200		E59	.	.	<200
	A8C	<200	3200	800		E8B	.	.	<200
	A94	<200	3200	1600		EC3	.	.	<200
	BFD	.	.	400		F04	<200	<200	<200
	C1F	<200	1600	1600 *		F91	<200	<200	<200
	C2A	.	.	6400		FFC	.	.	<200
	C80	<200	3200	800					
	CDE	.	.	1600					
	E73	.	.	3200					
	EF2	<200	6400	1600					
	F85	<200	6400	1600					
	F8B	<200	3200	800					

= No Sample. * = Sample collected on SD 207, prior to rabbit being removed from study for humane reasons. Animal did not enter the challenge phase.

**Table 3 viruses-16-00538-t003:** Geometric mean of serological response from the rBaculovirus-derived RHDV2 VP60 ELISA for the rabbits prior to challenge with RHDV2.

	Study Day		
Group	0	91	226 or 227
1—Vaccinate	107	3940	1600
2—Placebo	100	100	100

**Table 4 viruses-16-00538-t004:** Serological response from the rBaculovirus-derived RHDV2 VP60 ELISA for the rabbits monitored and boosted 12 months post-second vaccination (MPV).

Rabbit ID	Treatment Group	7 MPV2	9 MPV2	12 MPV2/0 DPB	14 DPB	28 DPB
202	Control	<200	<200	<200	1600	1600
677		<200	<200	<200	<200	<200
926		<200	<200	<200	200	400
4F4		<200	<200	<200	1600	3200
C16		<200	<200	<200	200	3200
**Geo Mean ***		**200**	**200**	**200**	**460**	**1056**
927	Vaccinate	1600	1600	800	12,800	12,800
34F		3200	1600	1600	≥25,600	≥25,600
AA3		1600	800	800	≥25,600	≥25,600
FEE		1600	1600	800	12,800	6400
**Geo Mean ***		**1903**	**1345**	**951**	**18,102**	**15,222**

* To calculate GeoMean, the < and > signs were removed. GeoMeans of 200 are considered negative.

## Data Availability

All available data for this study are included in the publication.

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
