# Peer review of "Vaccination against Rabbit Hemorrhagic Disease Virus 2 (RHDV2) Using a Baculovirus Recombinant Vaccine Provides Durable Immunity in Rabbits"

_viruses, 2024, doi:10.3390/v16040538_

Round 1

Reviewer 1 Report

Comments and Suggestions for Authors

This is an important paper that addresses critical questions about the rabbit hemorrhagic disease vaccine that has been in use for the last couple of years. The manuscript is well-written. Few comments are given below to clarify some of the results presented.

Abstract: Line 12 Reword this sentence for example "spread rapidly once introduced in a rabbit population" instead of "between infected individuals"

Section 2-Line 143 Please provide information about the primary antibody used (monoclonal/polyclonal/in house or commercially available/source if applicable).

Section 3-Line 163-171 Clarify this section. Should state more clearly that 18/19 nonvaccinated animals succumbed to infection and 24 of 25 animals that were subclinical were vaccinated animals.  L 170-171. Please clarify "generlized systemic hemorrhages in which multiple organs were involved" Please list main organs involved.

Table 1:  F04 is listed as vaccinated, but in Table 2 F04 is listed in placebo group.

Table 1:  354 is listed as not vaccinated, but in Table 2 354 is listed in vaccinated group.

Table 2: listed C1F, but C1F is not listed in Table 1 for vaccinated animals.

Section 3.3 Serology (lines 182-202)

Would have been nice to see titers at 21 DPV – would reflect titer after single vaccination. Unless there was no seroconversion after single vaccination

Section 3.3 Line 190 – the authors refer to table 3 to discuss the geometric mean titer of vaccinated animals.  It is more interesting that 6 of the 10 rabbits had a four-fold decrease in titer from 91 to 226/227 SD, 3 of 10 had a 2-fold decrease, and 1 had no decrease in titer.

Discussion Line 237 Eravac is not a multivalent vaccine as is indicated.

Section 4. L 253 In discussion – it gets a little confusing when authors comparing a previous study that they did with this current study. Would be helpful if they moved the reference (22) from line 256 to just after “prior study” in line 253. At end of line 263, would be nice to add “In current study” importantly . . .

Reviewer 2 Report

Comments and Suggestions for Authors

The reviewed work examines the duration of immunity provided by vaccination with the commercial vaccine 6 months following the treetment. Challenge study demonstrate the complete protection after two dose of adjuvanted with aluminum hydroxide vaccine.

Some notes to the text are listed below:

Line 214  Table 3:

      Geometric mean of serological response from the rBaculovirus derived RHDV2 VP60 ELISA for the rabbits challenged with RHDV2.

It's better to write:

Geometric mean of serological response from the rBaculovirus derived RHDV2 VP60 ELISA for the rabbits prior to challeng with RHDV2.

 Line 218 Table 4:

      In the lines “Geo Mean*” the dot and number after it are not needed, since the measurement accuracy is below than 1

Reviewer 3 Report

Comments and Suggestions for Authors

The authors reported that the recombinant vaccine for rabbit heamorrhagic disease virus 2 (RHDV2) infection protented rabbits from lethal infection six months following vaccination. The present manuscript is a follow up of their previous report. Prevention of RHDV2 infection is critical worldwide. The results in the present study will provide a scientific evidence for the recombinant vaccine use. Expreimental design was scientific and the manuscript is written amost clearly. Minor revisions are required as follows.

L58-60, 90-91:The sequence in the reference 22 was a base sequence, not amino acid sequence, and the sequence was indicated to be RHDV1, not RHDV2, by BLAST search. Differences between RHDV1 VP60 and RHDV2 VP60 should be discribed in Introduction.

L68: Grounds for using '55' rabbits should be described.

L68: A scientif name of New Zealand white rabbits should be described.

L80: Why were'45' rabbits trnasported instead of '55'? A reason for reducing the number of rabbits should be described.

L96: How much recombinant protein was included in a 0.5mL dose? The amount of recombinant vaccine shoud be described such as a protein amount for vaccination.

L101-107: Virus amount included in the homogenate should be described based on quantitative RT-PCR.

L166: Temperature shoud be shown in Celsius, not Fahrenheit.

Tables 1, 2 and 4: It is better to show a lookup table for rabbit Id, Trovan ID and sample ID. Table 4: Add explanations for abbreviations (MPV and DPB).

Reviewer 4 Report

Comments and Suggestions for Authors

The manuscript submitted by Angela M Bosco-Lauth et al, entitled "Vaccination against rabbit hemorrhagic disease virus 2 using a baculovirus recombinant vaccine provides durable immunity in rabbits", describes complementary work to that published in 2022 on a subunit vaccine (RHDVG1.2 VP60 protein produced by a baculovirus in SF9 cells) already registered in the USA (Medgene Platform). The aim of this study was to verify vaccine efficacy 6 months post-vaccination, as well as the persistence of the specific humoral response 12 months post-vaccination and following a booster injection at 1 year. The results are convincing, well described and illustrated. Experimental protocols are well explained. However, information on vaccine preparation is very brief (as was the case for the 2022 article), and vaccine doses are not documented (only the volumes injected are available). These points should be improved. Other elements need to be discussed (see questions and suggestions below).

Title: add RHDVGI.2 in brackets after rabbit hemorrhagic disease virus 2
Introduction:
Line 24: replace calicivirus with Caliciviridae
Line 33: remove "domestic and…", and replace "cuniculi" with "cuniculus".
Line 46: replace "rabbits" with lagomorphs (also includes hares)
Line 50: A third vaccine against RHDV2 (not including the NOBIVAC MYXO-RHD PLUS recombinant vaccine from MSD MSD, which provides simultaneous protection against myxomatosis, RHDV and RHDV2) is available in Europe: YURVAC-RHD from Hipra. It consists of VP60 proteins produced in vitro.
Line 51: Indicate that published vaccine candidates target RHDV and/or RHDV2. The bibliographical references cited at this level should be supplemented by the following (the first to my knowledge demonstrating the vaccine protection obtained with VLPs, and those describing the construction of MYXV-RHD recombinant viruses and the vaccine protection obtained):

S Laurent, J F Vautherot, M F Madelaine, G Le Gall, D Rasschaert. Recombinant rabbit hemorrhagic disease virus capsid protein expressed in baculovirus self-assembles into viruslike particles and induces protection. Journal of Virology, 1994, Vol. 68, No. 10. Doi : 10.1128/jvi.68.10.6794-6798.1994

S Bertagnoli, J Gelfi, G Le Gall, E Boilletot, J F Vautherot, D Rasschaert, S Laurent, F Petit, C Boucraut-Baralon, A Milon. Protection against myxomatosis and rabbit viral hemorrhagic disease with recombinant myxoma viruses expressing rabbit hemorrhagic disease virus capsid protein. Journal of Virology, 1996, Vol. 70, No. 8 doi :10.1128/jvi.70.8.5061-5066.1996

Barcena J, Morales M, Vazquez B, Boga JA, Parra F, Lucientes J, Pages-Mante A, Sanchez-Vizcano JM, Blasco R, Torres JM.Horizontal transmissible protection against myxomatosis and rabbit haemorrhagic disease using a recombinant myxoma virus. Journal of Virology, 2000;74:1114-23. doi :10.1128/jvi.74.3.1114-1123.2000

Spibey, N.; McCabe, V.J.; Greenwood, N.M.; Jack, S.C.; Sutton, D.; van der Waart, L. Novel bivalent vectored vaccine for control of myxomatosis and rabbit haemorrhagic disease. Vet. Rec. 2012, 170, 309.

Sylvia Reemers, Leon Peeters , Joyce van Schijndel , Beth Bruton , David Sutton , Leo van der Waart  and Saskia van de Zande. Novel Trivalent Vectored Vaccine for Control of Myxomatosis and Disease Caused by Classical and a New Genotype of Rabbit Haemorrhagic Disease Virus. Vaccines 2020, 8, 441; doi:10.3390/vaccines8030441

Materials and methods:

Line 88 (vaccine preparation): give more information on the baculovirus system used, the RHDV2 strain used to clone the VP60 gene, a brief method for preparing antigens. Is the prepared VP60 self-assembled in VLP?

Line 96: How much antigen was administered to the vaccinated rabbits (we only know the volume injected)?

Line 118: have you quantified the challenge viral  preparation (by quantitative RT-PCR for example)?

Line 129: real -time Taqman RT-PCR (instead of real-time reverse transcriptase)

Results :

Line 166: express body temperature in international units (degrees Celcius)

Line 175: "All rabbits that succumbed to infection had RT-PCR....".

Line 203: figures should not be collected in a specific sub-chapter (see recommendations to authors)

Discussion:

Line 262: Gall et al (reference 29 cited on line 263) describe the detection of viral RNA by real-time RT-PCR for 15 weeks in various tissues and organs of convalescent rabbits, but cannot conclude that infectious viruses (not detected by this work) are present. The sentence should therefore be amended to specify that it concerns the persistence of viral RNA.
